# Design and Experimental Study on an Innovative UAV-LiDAR Topographic Mapping System for Precision Land Levelling

**Mengmeng Du** [1,*] **, Hanyuan Li** [1] **and Ali Roshanianfard** [2]

1   College of Agricultural Equipment Engineering, Henan University of Science and Technology, Luoyang 471003, China
2   Department of Agriculture and Natural Resources, University of Mohaghegh Ardabili, Ardabil 566199, Iran
*   Correspondence: dualmon.du@haust.edu.cn

**Abstract:** Topographic maps provide detailed information on variations in ground elevation, which is essential for precision farmland levelling. This paper reports the development and experimental study on an innovative approach of generating topographic maps at farmland-level with the advantages of high efficiency and simplicity of implementation. The experiment uses a low-altitude Unmanned Aerial Vehicle (UAV) as a platform and integrates Light Detection and Ranging (LiDAR) distance measurements with Post-Processing Kinematic Global Positioning System (PPK-GNSS) coordinates. A topographic mapping experiment was conducted over two fields in Henan Province, China, and primitive errors of the topographic surveying data were evaluated. The Root Mean Square Error (RMSE) between elevation data of the UAV-LiDAR topographic mapping system and ground truth data was calculated as 4.1 cm and 3.6 cm for Field 1 and Field 2, respectively, which proved the feasibility and high accuracy of the topographic mapping system. Furthermore, the accuracies of topographic maps generated using different geo-spatial interpolation models were also evaluated. The results showed that a TIN (Triangulated Irregular Network) interpolation model expressed the best performances for both Field 1 with sparse topographic surveying points, and Field 2 with relatively dense topographic surveying points, when compared with other interpolation models. Moreover, we concluded that as the spatial resolution of topographic surveying points is intensified from 5 m $\times$ 0.5 m to 2.5 m $\times$ 0.5 m, the accuracy of the topographic map based on the TIN model improves drastically from 7.7 cm to 4.6 cm. Cut-fill analysis was also implemented based on the topographic maps of the TIN interpolation model. The result indicated that the UAV-LiDAR topographic mapping system could be successfully used to generate topographic maps with high accuracy, which could provide instructive information for precision farmland levelling.

**Keywords:** topographic map; agricultural remote sensing; unmanned aerial vehicle; light detection and ranging; interpolation model

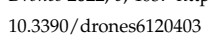

## 1. Introduction

Surface unevenness of farmlands has been pointed out as a major issue that affects agricultural drainage efficiency [1]. Hu et al. (2014) reported that over 20% of irrigation water could be wasted because of the rough land surface of paddy fields. Rickman (2002) concluded that unleveled rice fields suffered about 24% yield loss on average when compared to effectively leveled ones of the same rice variety and the same fertilizer input [2,3]. As precision farmland levelling facilitates agricultural drainage, improves crop establishment, and increases crop yield, many researchers both in academia and industry have proposed different methodologies for removing mounds and puddles in farmland [4,5]. Generally, two major precision farmland levelling techniques can be enumerated: laser farmland levelling and GNSS (Global Navigation Satellite System) farmland levelling [6]. An accurate topographic map is essential to precision farmland levelling; this map type refers to a 2D (two-dimensional) graphic representation of a terrestrial land surface feature

using contour lines, hypsometric tints, and relief shading [7–9]. As for the laser leveler, the preset height of the laser transmitter directly determines the cutting depth of each point, i.e., cut-fill ratios. It is almost impossible to set an appropriate height efficiently and accurately without referring to an accurate topographic map [10]. On the other hand, the GNSS leveler measures and adjusts the height of the scraper by comparing the current height of the GNSS receiver with the reference height plane to complete the earth-moving operation. It also needs an accurate topographic map to form the reference height plane. Thus, the accuracy, efficiency, and energy consumption of precision farmland levelling are in high accordance with the delicacy of farmland topographic maps [10].

To generate topographic maps for civil construction surveys, urban ecology modeling, forest monitoring, etc., new technologies including terrestrial laser scanning, aerial photogrammetry, and airborne laser scanning were recently utilized [11–18], while theodolite, total station, and handheld RTK (Real-Time Kinematic)-GNSS modules remain the conventional and primary tools used in common topographic surveys [19–21]. Topographic survey using a handheld GNSS module or tripod-based total station has low efficiency and cannot be applied to large-area operations [22]. On the other hand, topographic survey via remote sensing covers large areas and is capable of obtaining adequate data, in spite of the high cost and complicated calibration procedures [23]. Corsini et al. (2013) monitored and mapped a slow-moving compound rockslide using an integration of an airborne laser scanner, terrestrial laser scanner, and automated total station, which quantified slope movement in the order of centimeters to a few decimeters [24]. Rodriguez et al. (2017) evaluated a mobile LiDAR (Light Detection and Ranging) system mounted on a car to develop an architectural analysis by generating a 3D point cloud [25]. Terrone et al. (2021) identified manmade landforms and assessed the morphological evolution of the city of Genoa by coupling historical maps and LiDAR data [26]. There are also plenty of research works and products providing a DSM (Digital Surface Model) based on airborne photogrammetry or satellite stereo-imagery [27–33]. However, processing aerial images to generate photogrammetric DSMs usually needs strict camera calibration and a large number of spatially well-distributed GCPs (Ground Control Points), which in photogrammetry or the computer vision domain refers to such features that are easily recognizable and distinguishable in both the real world and the images. The spatial resolution, as well as the accuracy, of such topographic maps usually reaches several decimeters to tens of meters [34], which cannot meet the high requirements of topographic maps with centimeter-level accuracy for precision farmland levelling.

Therefore, this study developed and experimentally evaluated an innovative topographic mapping system for precision farmland levelling that was based on a low-altitude UAV (Unmanned Aerial Vehicle). The UAV was equipped with a LiDAR device and PPK (Post-Processing Kinematic)-GNSS modules, which are utilized to conduct topographic survey in a simple and totally autonomous manner. This research integrates multi-source remote sensing data on board the UAV platform, including LiDAR distance measurements, attitude information (pitch and roll) of the UAV's flight controller, and PPK-GNSS positioning data (latitude, longitude, altitude). The ultimate objective is to generate topographic maps of farmlands with centimeter-level accuracy efficiently, both in terms of time and cost, using geo-spatial interpolation models.

## 2. Materials and Methods

### 2.1. Experiment Equipment

A farmland topographic mapping experiment was established over two fields located in Kaifeng City, Henan Province, China. Each field accounts for about 1500 m$^2$ and 1200 m$^2$, respectively, shown in Figure 1 as Field 1 and Field 2. The previous crop was maize. Weeds in the fallow fields were manually removed prior to the experiment so that there were no significant foreign attachments on the ground surface.

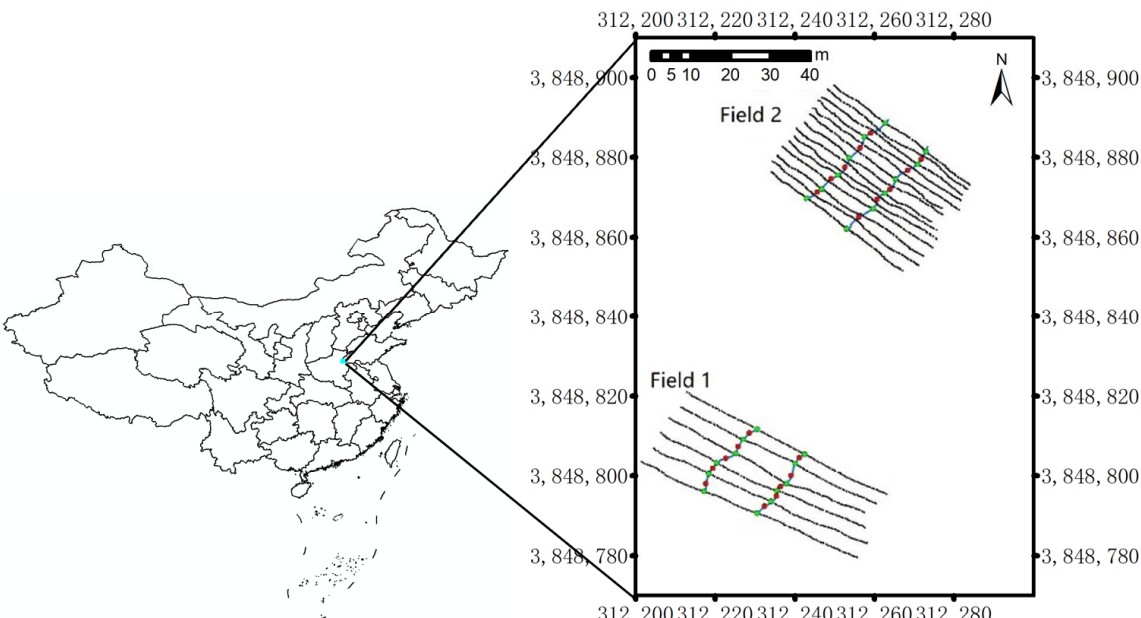

**Figure 1.** Flight path of UAV-LiDAR topographic surveying experiment and locations of accuracy validating points. Note: The black lines, green dots, and red dots indicate the flight trajectories of the UAV, the sampling points of handheld PPK-GNSS for evaluating the accuracy of ground surveying points of the UAV-LiDAR topographic surveying system, and the sampling points of handheld PPK-GNSS for evaluating the accuracy of topographic maps based on interpolation models.

A hexacopter was used as the UAV platform, shown in Figure 2a. A high-accuracy LiDAR distance measuring device (JENOPTIK, Jena, Germany) was used to measure the distance between the ground surface and the UAV-LiDAR topographic mapping system, which was rigidly fixed under the UAV platform pointing vertically downwards. Two GNSS modules were used to calculate the 3D positioning coordinates at the frequency of 10 Hz of the UAV-LiDAR topographic mapping system in the PPK manner. One GNSS module was installed on the top of the UAV as a rover receiver, while the other was fixed nearby the field as a base receiver, shown in Figure 2b. The parameters of the equipment of the UAV-LiDAR topographic mapping system are listed in Table 1.

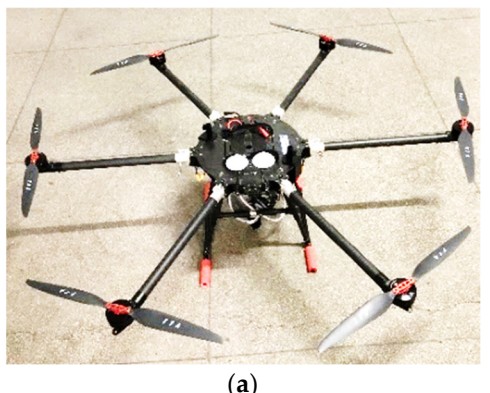

(**a**)

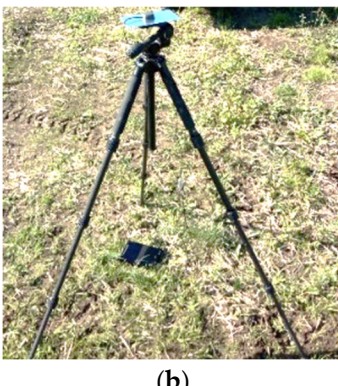

(**b**)

**Figure 2.** Equipment of the topographic surveying experiment. (**a**) UAV platform; (**b**) Base receiver of PPK-GNSS module.

**Table 1.** Parameters of the equipment of the UAV-LiDAR topographic mapping system.

| Equipment | Items | Values |
|---|---|---|
| UAV platform | Overall size (mm) | $\varnothing$1290 $\times$ 650 |
| | Motor type | TTA6215 |
| | Standard take-off weight (kg) | 23 |
| | Flight controller | Pixhawk |
| | Flight altitude (m) | 30 |
| GNSS module | GNSS receivers | U-blox NEO-M8T |
| | Processing unit | Intel Edison |
| | Overall size (mm) | 45.5 $\times$ 27 $\times$ 9.2 |
| | Weight (g) | 14 |
| | Supply voltage (V) | 4.75 to 5.5 DC |
| | Power consumption (W) | <1 |
| | Signals | GNSS, GLONASS, BDS |
| | Out frequency (Hz) | 10 |
| | Antenna | Tallysman TW4721 |
| LiDAR | Overall size (mm) | 136 $\times$ 104 $\times$ 57 |
| | Weight (g) | 800 |
| | Measuring resolution (mm) | 1 |
| | Output frequency (Hz) | 2000 |
| | Measuring range (m) | 300 |
| | Beam divergence (milliradians) | 1.7 |
| | Measuring laser wavelength (nm) | 905 |
| | Supply voltage (V) | 10 to 30 DC |
| | Power consumption (W) | <5 |

Autonomous UAV flight was conducted on 24 October 2020, using flight paths designed beforehand at the speed of about 5 m/s and altitude of about 30 m above ground level. The flight speed was determined with consideration of the efficiency and attitude stability of the UAV platform. Low flight speed results in low efficiency, and when the flight speed increases, the pitch angle of UAV body correspondingly changes violently, which affects the stability of the topographic surveying system and accuracy of laser ranging data. From the flight speed and positioning frequency of the GNSS module, in-track intervals of the UAV-LiDAR topographic surveying points could be determined as 0.5 m. The cross-track interval of Field 1 and Field 2 was set at about 5 m and 2.5 m, respectively, to cover the whole field in each flight, according to the endurance limitation of common civilian UAVs. The influence of spatial resolution of the topographic surveying points on the accuracy of topographic maps by using interpolation models was studied; this is jointly determined by the cross-track interval and the in-track interval. As the in-track interval was already far less than the cross-track interval, the merits of further reducing the flight speed for acquiring denser topographic surveying points was not discussed. Therefore, the spatial resolution of the topographic surveying data for Field 1 and Field 2 was calculated as about 5 m $\times$ 0.5 m and 2.5 m $\times$ 0.5 m, respectively. The trajectories of two autonomous flights over the experimental fields are shown in Figure 1. Because the altitude of the UAV-LiDAR topographic surveying system affects the accuracy of LiDAR's distance measurement, it should be set as low as possible. In this study, the altitude was set to 30 m above ground level to conduct accurate topographic surveying and to simultaneously capture aerial images for further study of data integration.

### 2.2. Acquiring PPK-GNSS Coordinates

Theoretically, a standalone GNSS receiver usually has a positioning accuracy varying from submeter to several meters [35,36]. On the other hand, an RTK-GNSS module uses a network of ground or virtual reference stations to rectify the GNSS rover receiver's positioning data, and its positioning accuracy could be significantly improved up to about 2~5 cm. However, as the RTK-GNSS device is embedded with communication modules

for data transmission via radio signal or cellular network, it is always too large in size and/or weight to be installed upon small-sized UAV. Therefore, in this study two identical lightweight PPK-GNSS modules were used to collect high-accuracy positioning data. After the completion of a UAV topographic mapping flight, GNSS positioning data of the rover receiver and the base receiver were acquired; we could dynamically rectify the GNSS rover receiver's positioning data in the post-processing style with the reference of the GNSS base receiver's positioning data in the following manner.

Firstly, both the GNSS base and rover receiver were held stationary before the UAV platform took off for over 30 min consecutively, and therefore 18,000 sets of the effective static positioning data of each GNSS module at the frequency of 10 Hz were obtained. By using an arithmetic averaging method, the reference coordinate $(x_r, y_r, h_r)$ for PPK-GNSS algorithm was calculated from the GNSS base receiver's positioning data $(x_{basei}, y_{basei}, h_{basei})$ as (3,848,750.598 E, 312,414.385 N, 63.754 m), according to Equations (1)–(3).

$$x_r = \sum_{i=1}^{18000} \frac{x_{basei}}{18000} \tag{1}$$

$$y_r = \sum_{i=1}^{18000} \frac{y_{basei}}{18000} \tag{2}$$

$$h_r = \sum_{i=1}^{18000} \frac{h_{basei}}{18000} \tag{3}$$

where $x_r$, $y_r$, and $h_r$ refer to easting, northing, and altitude of the reference coordinate for the PPK algorithm, while $x_{basei}$, $y_{basei}$, and $h_{basei}$ refer to easting, northing, and altitude of the positioning data of the GNSS base receiver, respectively.

Subsequently, the positioning data of the GNSS rover receiver $(x_{rovi}, y_{rovi}, h_{rovi})$ and the base receiver over the same period were processed in the RTKLIB development environment using the PPK-GNSS algorithm, where $x_{rovi}$, $y_{rovi}$, and $h_{rovi}$ refer to easting, northing, and altitude of positioning data of the GNSS rover receiver, respectively. The PPK-GNSS algorithm calculates position deviations between each positioning datum of the GNSS base receiver $(x_{basei}, y_{basei}, h_{basei})$ and the reference coordinate $(x_r, y_r, h_r)$ based on carrier phase analysis. It then uses the deviation information to rectify the corresponsive positioning data of GNSS rover receiver $(x_{rovi}, y_{rovi}, h_{rovi})$. Thus, PPK-GNSS positioning data were acquired as $(x_{GPSi}, y_{GPSi}, h_{GPSi})$.

Finally, the GNSS base receiver was kept at the same location, and the GNSS rover receiver on board the UAV platform was used to conduct the topographic mapping experiment. The UAV flight lasted for about 12 min for each field, and in total 675 and 1626 sets of PPK-GNSS positioning data were obtained for Field 1 and Field 2, respectively, after removing noise data during take-off, landing, and turn-around. The horizontal positioning coordinates of PPK-GNSS data were previously given in Figure 1, while the altitudes of PPK-GNSS data are shown in Figure 3. In Figure 3, variations in the UAV's flight altitude can be observed from 90 m to 92.5 m, which is caused by its intrinsic aerodynamic factors and the influence of turbulence. Upon completion of the topographical mapping flight, the GNSS rover receiver was removed from the UAV platform and fixed onto an aluminum plate. Subsequently, it was dragged manually over the field ground to cross the UAV's flight paths, shown in Figure 1 as green lines, in order to acquire PPK-GNSS data for accuracy evaluation.

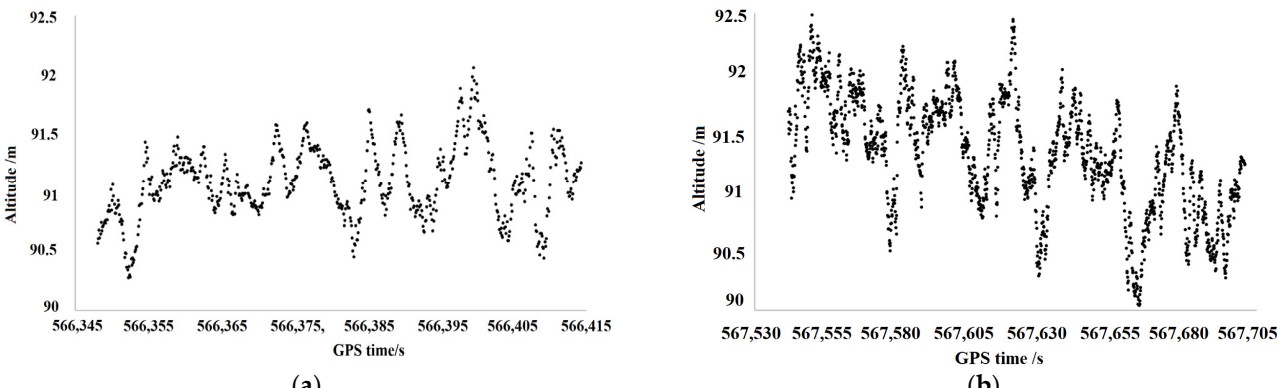

**Figure 3.** Altitudes of PPK-GNSS positioning data on board the UAV platform. (**a**) Field 1; (**b**) Field 2.

### 2.3. Rectifying LiDAR Distance Measurements Using UAV Attitude Data

LiDAR is based on the time-of-flight principle and has a measuring range up to 300 m and a 1 mm measuring resolution at the frequency of 2000 Hz [37–40]. The laser beam divergence of the LiDAR distance measurement device was 1.7 milliradians, and the spatial coverage of each measurement was about 60 mm × 20 mm from a distance of about 30 m. An onboard computer was connected with the LiDAR through a universal serial bus cable; it saves each distance measurement on board in real time. As the output frequency of the PPK-GNSS's positioning data was 10 Hz, a mean filter with a 200 step size was applied to raw LiDAR distance measurements. Thus, the output frequency of the LiDAR distance measurements was altered from 2000 Hz to 10 Hz in order to align multi-sources remote sensing data according to time sequences and also to improve the LiDAR distance measuring accuracy in the meantime. Due to its aerodynamic characteristics, the UAV body is apt to tilt irregularly during cruise flight [41]. Because electronical gimbals cannot meet the real-time stabilizing requirements of this study, the LiDAR was rigidly fixed below the UAV platform and the UAV's attitude information was used to compensate for such influences as vibration and air disturbance during flight.

Attitude of a UAV is usually described by means of Euler angles defined as pitch ($\theta$), roll ($\phi$), and yaw ($\psi$) [42–44]. In this study, we utilized an extended Kalman filter to integrate data from multiple sensors of the MEMS (Micro-Electro-Mechanical System) gyroscope, accelerometer, and magnetometer for attitude estimation. A gyroscope is an inertial sensor for measuring orientation based on angular momentum principles, and tri-axial angles $\left[\theta_{gyro} \, \phi_{gyro} \, \psi_{gyro}\right]^T$ can be acquired by integral operation, as expressed in Equation (4). However, due to temperature variations in the gyroscope, errors accumulate along with time and the accuracy of the MEMS gyroscope's attitude data will be compromised. On the other hand, an accelerometer was used to measure the UAV's orientation based on the trigonometric functions of the acceleration of gravity components in each axial, expressed in Equation (5) and Equation (6). A MEMS magnetometer is also used for precisely calculating change of the UAV's heading yaw based on magnetic intensity components in each axial $\left[m_x, m_y, m_z\right]^T$, in combination with measurements from the gyroscope and accelerometer [45,46].

$$\left[\theta_{gyro} \, \phi_{gyro} \, \psi_{gyro}\right]^T = \int \left[\omega_x \, \omega_y \, \omega_z\right]^T dt \qquad (4)$$

where $\theta_{gyro}$, $\phi_{gyro}$, and $\psi_{gyro}$ are pitch, roll, and yaw of the UAV platform from the gyroscope, respectively, while $\omega_x$, $\omega_y$, and $\omega_z$ are the raw rotation rates (degree/s) of each axial from the gyroscope.

$$\theta_{acce} = -arcsin\left(\frac{g_x}{\sqrt{g_x^2 + g_y^2 + g_z^2}}\right) \qquad (5)$$

$$\phi_{acce} = arctan\left(\frac{g_y}{g_z}\right) \tag{6}$$

where $\theta_{acce}$ and $\phi_{acce}$ are pitch and roll of the UAV from the accelerometer, respectively, while $g_x$, $g_y$, and $g_z$ are the acceleration components readings in each axial from the accelerometer.

Therefore, in order to further improve the accuracy of LiDAR distance measurements, attitude data (pitch $\theta_i$ and roll $\varphi_i$) of the UAV platform were used to obtain the nadir distance, according to Equation (7). Nadir is defined as the angle that points directly downward, or 0°, from the luminaire, as shown in Figure 4. Variations in the nadir distance between the LiDAR device and ground surface during the UAV-LiDAR topographic surveying process are shown in Figure 5. The nadir distance between the UAV-LiDAR system and ground surface varied from 25.5 m to 28.5 m for both Field 1 and Field 2. The variations in the nadir distance between the UAV-LiDAR topographic mapping system and ground surface arise from many factors, such as the ever-changing altitude of the UAV platform during flight and the field terrain itself.

$$d_{ni} = d_i \times \cos\theta_i \times \cos\varphi_i \tag{7}$$

where $d_{ni}$ and $d_i$ are the nadir distance and the actually measured distance between the LiDAR device and ground surface, respectively, while $\theta_i$ and $\varphi_i$ are the attitude angle of pitch and roll of the UAV flight controller, respectively.

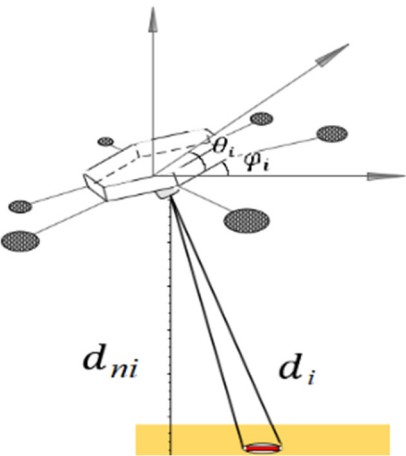

**Figure 4.** Scheme for rectifying LiDAR distance measurements using UAV attitude information.

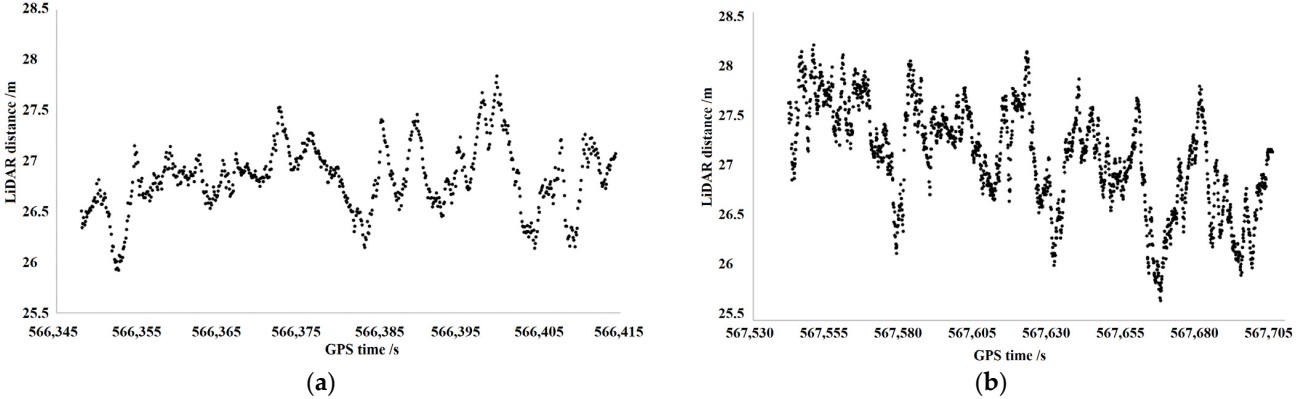

(**a**)

(**b**)

**Figure 5.** Nadir distance between the UAV-LiDAR system and ground surface. (**a**) Field 1; (**b**) Field 2.

### 2.4. Acquiring 3D Coordinates of Ground Surveying Points

The scheme for calculating the elevation of ground surveying points ($h_i$) is elaborated in Figure 6 and could be calculated according to Equation (8). Subsequently, by replac-

ing $h_{GPSi}$ with $h_i$, 3D coordinates of the ground surveying points could be obtained as $(x_{GPSi}, y_{GPSi}, h_i)$.

$$h_i = h_{GPSi} - h_{fix} - d_{ni} \tag{8}$$

where $h_i$, $h_{GPSi}$, $h_{fix}$, and $d_{ni}$ are the elevation of ground surveying points, the altitude of the UAV's PPK-GNSS coordinates, the installation height difference between the GNSS rover receiver and the LiDAR device (0.66 m), and the nadir distance between the UAV-LiDAR system and ground surface, respectively.

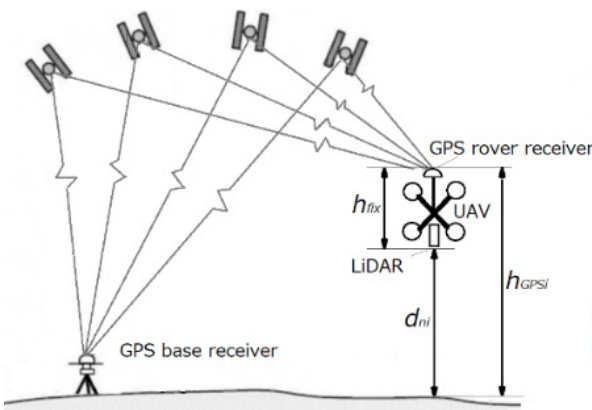

**Figure 6.** Scheme for calculating elevations of ground surveying points.

*2.5. Generating Topographic Maps Based on Interpolation Models*

In this study, 675 and 1626 sets of effective topographic mapping data for Field 1 and Field 2, respectively, were used to generate topographic maps based on different interpolation models. In geo-statistics, interpolation is a method for approximating and assigning new data for locations where no samples have been taken within the range of a discrete set of known data points. There are various interpolation models such as TIN (Triangulated Irregular Network), IDW (Inverse Distance Weighting), Kriging, and natural neighbor, etc. [47–50]; however, it remains unclear as to which model is the most suitable for farmland terrain research purposes.

TIN is a vector-based interpolation model that forms a network of triangles of irregular size and shape. TIN's triangular linear interpolation process begins with determining the triangular to which the interpolation point belongs, then calculating the parameters of a, b, and c for the plane equations of each triangular using the 3D coordinates of three vertices, expressed as Equation (9). The elevation data of every interpolation point within the triangular could be acquired according to Equation (10).

$$\begin{bmatrix} z_1 \\ z_2 \\ z_3 \end{bmatrix} = \begin{bmatrix} 1, & x_1, y_1 \\ 1, & x_2, y_2 \\ 1, & x_3, y_3 \end{bmatrix} \begin{bmatrix} a \\ b \\ c \end{bmatrix} \tag{9}$$

$$z = a + bx + cy \tag{10}$$

where $a$, $b$, and $c$ are the parameters for determining the plane equation, while $x_i$, $y_i$, and $z_i$ ($I$ = 1, 2, and 3) are the 3D coordinates (easting, northing, and elevation) of each of the vertices of the triangular, and $x$, $y$, and $z$ are the 3D coordinates of each interpolating point, respectively.

The IDW interpolation model assigns a weighted value of serval neighboring points by using an inverse distance weighted technique. The weight parameters could be determined from the mathematical model expressed as Equations (11)–(13). The natural neighbor inter-

polation model is similar to the IDW model, except that it determines weight parameters by calculating proportionate areas instead of inversed distances.

$$w_i = \frac{d_i^{-p}}{\sum_{i=1}^{n_{idw}} d_i^{-p}} \tag{11}$$

$$d_i = \sqrt{(x - x_i)^2 + (y - y_i)^2} \tag{12}$$

$$\hat{z}(x, y) = \sum_{i=1}^{n_{idw}} w_i z(x_i, y_i) \tag{13}$$

where $w_i$, $d_i$, $p$, and $n_{idw}$ are the weighting factor of each measured points, distance of each measured point to interpolating point, power parameter, and number of measured points included in the IDW model, respectively, while $(x, y)$ and $(x_i, y_i)$ are easting and northing coordinates of the interpolating point and each measured point, respectively; $\hat{z}(x, y)$ and $z(x_i, y_i)$ are the estimated ground elevation of the interpolating point and each measured point's ground elevation, respectively.

As mentioned above, TIN, IDW, and natural neighbor interpolation models use the surrounding points' z–values for estimating the z–value for each interpolating point. However, the Kriging interpolation model incorporates geostatistical relationships among the measured points. Thus, in contrast to the IDW model, the weight parameter for each surrounding measured point in the Kriging interpolation model is not only determined by the inversed distance, but also influenced by the overall spatial autocorrelation of the measured points, expressed as Equation (14).

$$\hat{z}(x, y) = \sum_{i=1}^{n_{kri}} w_i z(x_i, y_i) \tag{14}$$

where $\hat{z}(x, y)$, $n_{kri}$, $w_i$, and $z(x_i, y_i)$ are the estimated z-value of an interpolating point, number of measured points included in the Kriging model, weighting factor of each measured points, and each measured point's z-value, respectively.

## 3. Results and Discussion

### 3.1. Evaluating the Accuracy of a UAV-LiDAR Topographic Mapping System

Horizontal positioning data of the stationary GNSS base receiver in standalone and PPK mode is shown in Figure 7a,b, respectively, from which the horizontal positioning accuracy can be determined as about 3 m and 2 cm. The vertical positioning data varied within the range of 66.180 m to and 66.210 m, with an accuracy of about 3 cm, shown in Figure 7c. Therefore, Figure 7 clearly indicates that PPK-GNSS positioning data have high accuracy and are capable of precisely measuring the 3D coordinates of the UAV-LiDAR topographic mapping system.

Based on the 3D coordinates of each topographic surveying point, topographic maps indicating spatial variations in within-field ground elevation were generated by using ArcMap (ESRI Inc., Redlands, AB, Canada), shown in Figure 8. From the graduated symbols representing different levels of ground elevation, concave features (puddles) at each end of Field 1 can be visually specified and the general high-south-low-north terrain of Field 2, with ground elevations varying from 63.398 m to 63.761 m can be understood.

Accurate 3D coordinates (easting, northing, and elevation) of 12 discrete points distributed around each field were acquired for the accuracy evaluation; these overlapped (or fell very near to) the topographic mapping points. The geo-spatial coordinates of these points are listed in Table 2.

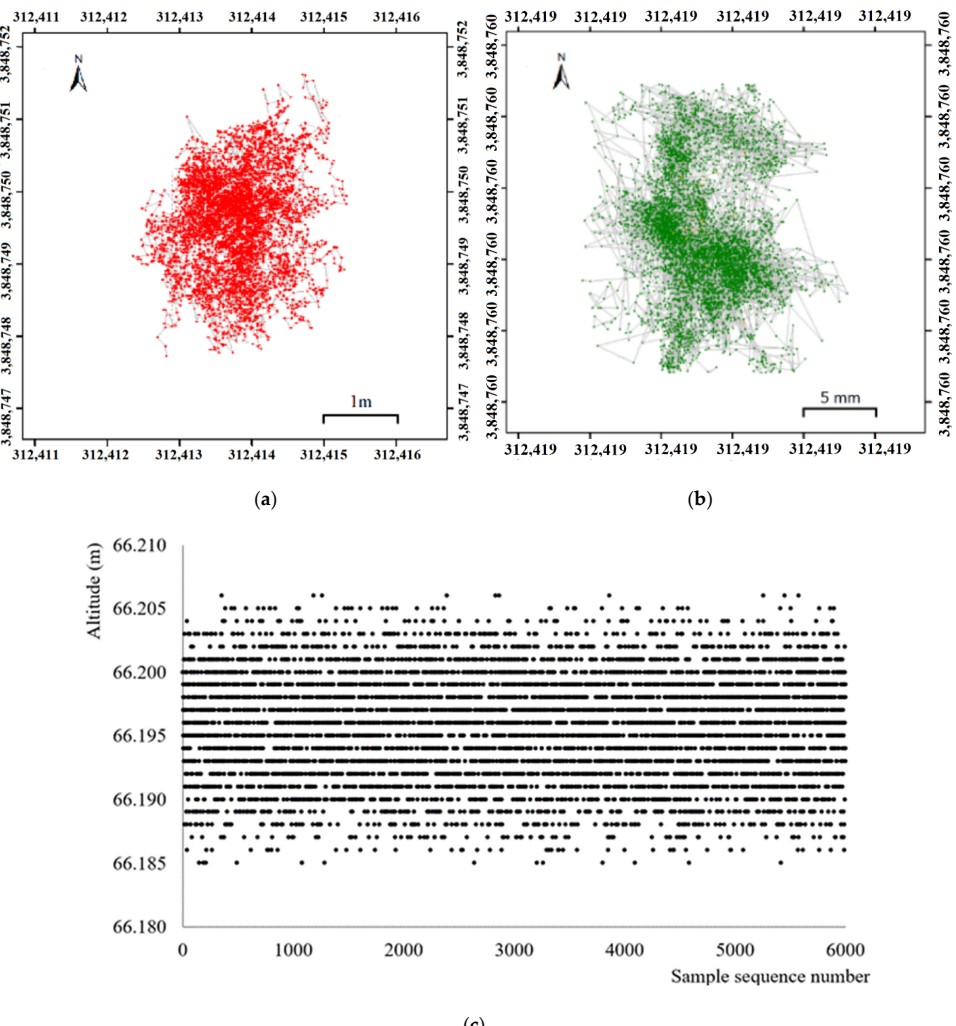

(a)

(b)

(c)

**Figure 7.** Positioning accuracy of standalone GNSS and PPK-GNSS. (**a**) Horizontal positioning accuracy of standalone GNSS; (**b**) Horizontal positioning accuracy of PPK-GNSS; (**c**) Vertical positioning accuracy of PPK-GNSS.

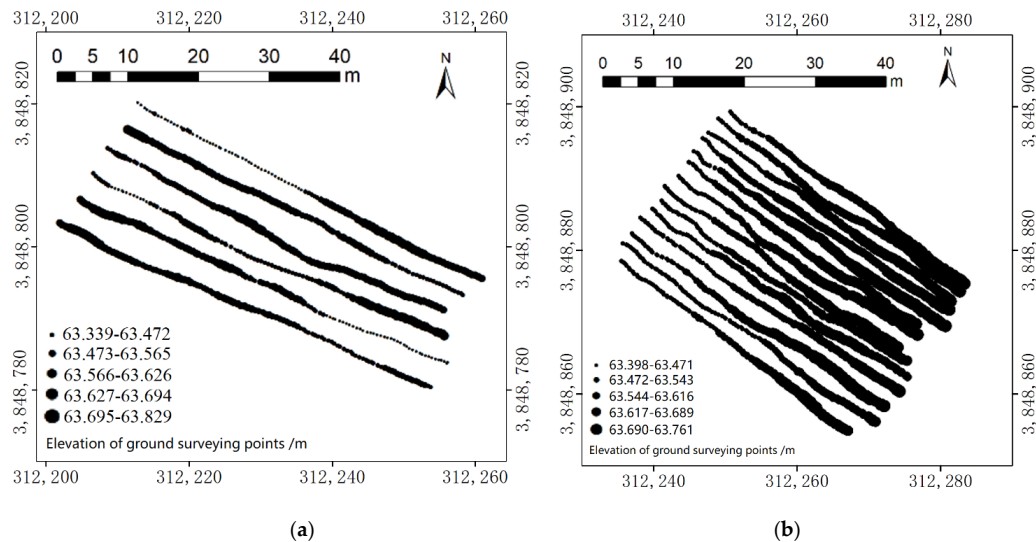

(a)

(b)

**Figure 8.** Graduated symbols represent elevations of ground surveying points. (**a**) Field 1; (**b**) Field 1.

**Table 2.** 3D coordinates of topographic mapping points measured by UAV-LiDAR topographic mapping system and PPK-GNSS module.

| Field NO. | Point NO. | Easting/m | Northing/m | Ground Elevation/m | |
|---|---|---|---|---|---|
| | | | | **UAV-LiDAR System** | **Handheld PPK-GNSS** |
| Field 1 | 1 | 312,217.07 | 3,848,795.93 | 63.658 | 63.616 |
| | 2 | 312,217.80 | 3,848,801.52 | 63.628 | 63.593 |
| | 3 | 312,219.74 | 3,848,803.24 | 63.64 | 63.655 |
| | 4 | 312,223.89 | 3,848,806.07 | 63.657 | 63.679 |
| | 5 | 312,226.45 | 3,848,809.20 | 63.649 | 63.607 |
| | 6 | 312,229.30 | 3,848,811.51 | 63.404 | 63.476 |
| | 7 | 312,229.85 | 3,848,790.93 | 63.605 | 63.565 |
| | 8 | 312,233.01 | 3,848,793.73 | 63.67 | 63.631 |
| | 9 | 312,234.60 | 3,848,796.05 | 63.514 | 63.568 |
| | 10 | 312,236.79 | 3,848,798.72 | 63.631 | 63.664 |
| | 11 | 312,239.10 | 3,848,802.99 | 63.572 | 63.529 |
| | 12 | 312,241.29 | 3,848,805.18 | 63.685 | 63.662 |
| Field 2 | 1 | 312,243.61 | 3,848,872.26 | 63.420 | 63.369 |
| | 2 | 312,247.92 | 3,848,874.06 | 63.522 | 63.508 |
| | 3 | 312,251.31 | 3,848,877.77 | 63.526 | 63.509 |
| | 4 | 312,254.22 | 3,848,881.96 | 63.531 | 63.596 |
| | 5 | 312,257.82 | 3,848,887.05 | 63.549 | 63.524 |
| | 6 | 312,263.32 | 3,848,890.26 | 63.532 | 63.584 |
| | 7 | 312,273.22 | 3,848,883.35 | 63.609 | 63.558 |
| | 8 | 312,270.82 | 3,848,880.46 | 63.635 | 63.696 |
| | 9 | 312,265.42 | 3,848,876.86 | 63.629 | 63.542 |
| | 10 | 312,262.92 | 3,848,873.27 | 63.641 | 63.657 |
| | 11 | 312,260.31 | 3,848,869.36 | 63.542 | 63.681 |
| | 12 | 312,253.82 | 3,848,864.66 | 63.535 | 63.578 |

The Root Mean Square Error (RMSE) between 12 pairs of ground elevation data of the handheld PPK-GNSS module and UAV-LiDAR topographic mapping system was calculated as 0.041 m and 0.036 m for Field 1 and Field 2, respectively. The RMSE for the two fields under study showed no significant distinction, although the track interval of the UAV-LiDAR topographic mapping system for the two fields was quite different (5 m for Field 1 and 2.5 m for Field 2). Since the ground elevation of the fields under study varied from 63.339 m to 63.829 m and 63.398 m to 63.761 m, respectively, the value of $RMSE_{UAV}$ showed that the topographic data of the UAV-LiDAR topographic mapping system are of very high precision.

### 3.2. Topographic Maps of Different Interpolation Models

In ArcMap software, topographic maps representing the ground elevation features of Field 1 and Field 2 using the TIN model were generated, shown in Figure 9. From the maps it may be concluded that ground elevation of Field 1 varies from 63.339 m to 63.829 m, and the distinguishing convex features (ridges) that are shown in white intertwine with concave features (puddles) that are shown in black, with no obvious trend of changing terrain. However, we can visualize a clear low-west-high-east terrain trend for Field 2, with the ground elevation changing from 63.398 m to 63.761 m.

The IDW interpolation result is shown in Figure 10, from which it can be seen more clearly that ridges and puddles spread all around Field 1 and the low-west-high-east terrain trend of Field 2 is also more obvious when compared with the topographic maps based on the TIN interpolation model.

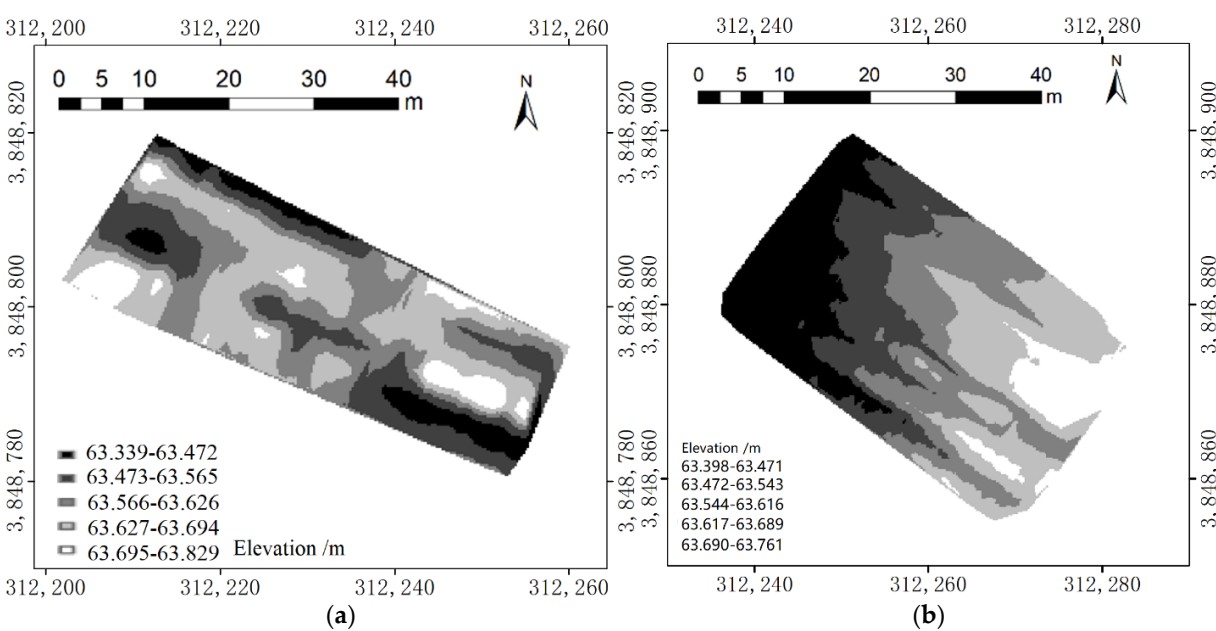

**Figure 9.** Topographic maps generated using the TIN interpolating model. (**a**) Field 1; (**b**) Field 2.

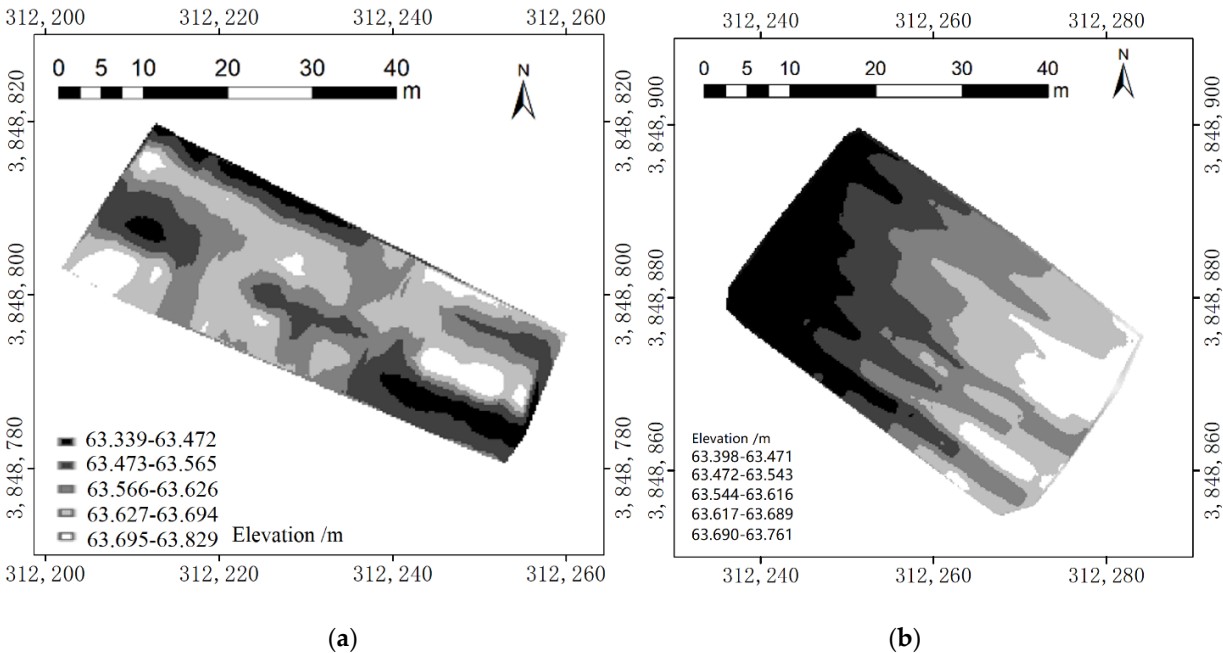

**Figure 10.** Topographic maps generated using the IDW interpolation model. (**a**) Field 1; (**b**) Field 2.

Topographic maps using the natural neighbor interpolation model, shown in Figure 11, were also generated in ArcMap software. From these, the similar terrain details of the two fields under study can be better observed in comparison to with the topographic maps generated by using the IDW interpolation model. Moreover, the topographic maps based on the natural neighbor interpolating model have a more drastic change in ground elevation.

In this study, an ordinary Kriging method with a spherical semi-variogram model was used in ArcMap software to generate topographic maps, shown in Figure 12, from which it may be concluded that the topographic maps based on the Kriging interpolation model are quite similar to the topographic maps based on the IDW model, and the former are featured with more smooth edges in terms of terrain change.

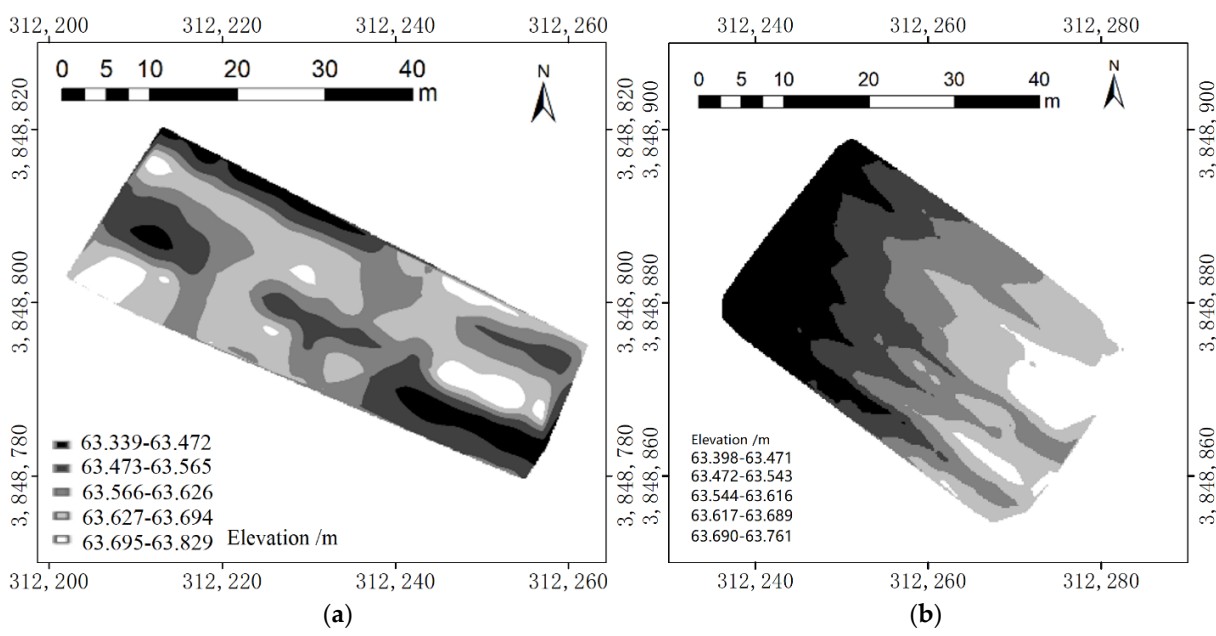

**Figure 11.** Topographic maps generated using the natural neighbor interpolation model. (**a**) Field 1;
(**b**) Field 2.

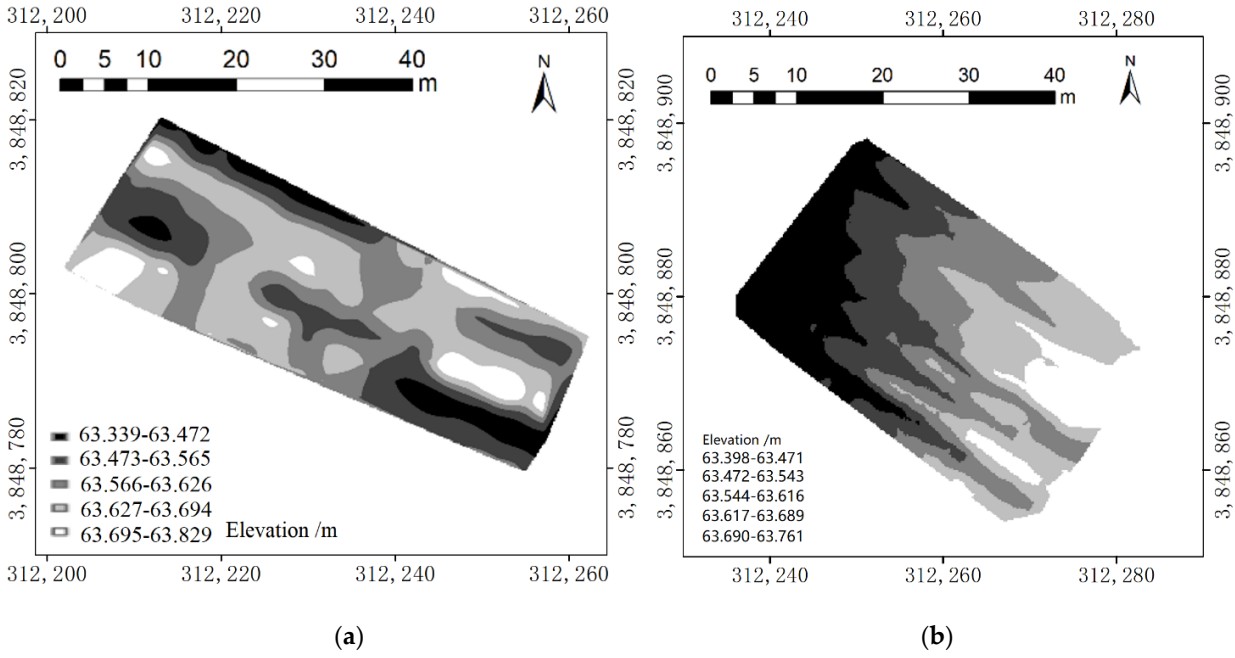

**Figure 12.** Topographic maps generated using the Kriging interpolation model. (**a**) Field 1; (**b**) Field 2.

### 3.3. Evaluating Accuracy of Topographic Maps

The accuracy of topographic maps generated using interpolation models depends
on two systems of error. One is the accuracy of each measured point, i.e., the primitive
error. The other adheres to a mathematical model of each specific interpolation model,
i.e., the interpolation error. In this study, the primitive error is indicated by the values of
RMSE between the ground elevation data of the handheld PPK-GNSS module and the
UAV-LiDAR topographic mapping system, which proved that the topographic data of the
UAV-LiDAR topographic mapping system are very accurate. Interpolation error is the
main source that contributes to the accuracy of topographic maps as the z–value of each
interpolation point is not only determined by the surrounding points' z–values, but also is
influenced by the neighboring points' spatial distribution.

In order to evaluate the accuracy of topographic maps generated using different interpolation models, accurate 3D coordinates of 10 discrete points distributed around each field from the trajectory of the handheld PPK-GNSS module were obtained. The spatial distribution of these evaluating points was shown as green dots in Figure 1, and the 3D coordinates were listed in Table 3. Furthermore, according to the easting and northing coordinates of the handheld PPK-GNSS data, the ground elevations of each point in the topographic maps of the different interpolation models were extracted in ArcMap software and listed in Table 3.

**Table 3.** 3D coordinates of sampled points acquired using a handheld PPK-GNSS module and different interpolation models.

| Field NO. | Point NO. | Easting/m | Northing/m | Ground Elevation/m | | | | |
|---|---|---|---|---|---|---|---|---|
| | | | | PPK-GNSS | TIN | IDW | Natural Neighbor | Kriging |
| Field 1 | 1 | 312,227.81 | 3,848,810.62 | 63.461 | 63.489 | 63.476 | 63.501 | 63.497 |
| | 2 | 312,225.17 | 3,848,807.27 | 63.587 | 63.657 | 63.655 | 63.654 | 63.654 |
| | 3 | 312,222.05 | 3,848,804.39 | 63.693 | 63.645 | 63.636 | 63.632 | 63.631 |
| | 4 | 312,240.04 | 3,848,804.39 | 63.618 | 63.637 | 63.644 | 63.644 | 63.644 |
| | 5 | 312,218.93 | 3,848,801.75 | 63.585 | 63.636 | 63.642 | 63.641 | 63.643 |
| | 6 | 312,238.12 | 3,848,800.07 | 63.656 | 63.619 | 63.629 | 63.617 | 63.627 |
| | 7 | 312,217.25 | 3,848,798.15 | 63.578 | 63.631 | 63.642 | 63.638 | 63.642 |
| | 8 | 312,235.48 | 3,848,797.43 | 63.462 | 63.570 | 63.579 | 63.575 | 63.577 |
| | 9 | 312,234.52 | 3,848,795.03 | 63.704 | 63.590 | 63.573 | 63.585 | 63.586 |
| | 10 | 312,231.64 | 3,848,792.63 | 63.779 | 63.639 | 63.636 | 63.633 | 63.634 |
| Field 2 | 1 | 312,259.81 | 3,848,887.79 | 63.589 | 63.546 | 63.543 | 63.546 | 63.546 |
| | 2 | 312,257.09 | 3,848,884.10 | 63.520 | 63.572 | 63.581 | 63.578 | 63.579 |
| | 3 | 312,272.05 | 3,848,881.38 | 63.676 | 63.626 | 63.623 | 63.625 | 63.624 |
| | 4 | 312,253.59 | 3,848,879.43 | 63.554 | 63.508 | 63.504 | 63.502 | 63.502 |
| | 5 | 312,268.75 | 3,848,878.85 | 63.683 | 63.636 | 63.626 | 63.634 | 63.633 |
| | 6 | 312,250.09 | 3,848,876.71 | 63.538 | 63.481 | 63.474 | 63.485 | 63.483 |
| | 7 | 312,264.47 | 3,848,874.19 | 63.670 | 63.651 | 63.651 | 63.652 | 63.652 |
| | 8 | 312,246.78 | 3,848,873.41 | 63.495 | 63.471 | 63.462 | 63.471 | 63.469 |
| | 9 | 312,261.36 | 3,848,871.66 | 63.519 | 63.578 | 63.594 | 63.586 | 63.588 |
| | 10 | 312,256.89 | 3,848,867.38 | 63.593 | 63.549 | 63.546 | 63.549 | 63.551 |

The RMSE between the ground elevation data of the handheld PPK-GNSS module and topographic maps of different interpolation models for Field 1 and Field 2 was calculated and listed in Table 4. From Table 4, it could be concluded that although there is no significant distinction between the primitive error of the UAV-LiDAR topographic mapping system for Field 1 and Field 2, the difference of interpolating errors among each interpolation model is obvious. The TIN interpolation model expressed the best performances for both Field 1 with sparse topographic surveying points, and Field 2 with relatively dense topographic surveying points, when compared with the other interpolation models including IDW, natural neighbor, and Kriging. Furthermore, as the spatial resolution of the topographic surveying points is improved from 5 m × 0.5 m to 2.5 m × 0.5 m, the interpolating error of the topographic maps based on the TIN model drops drastically from 0.077 m to 0.046 m.

**Table 4.** RMSE between ground elevation data of a handheld PPK-GNSS module and topographic maps of different interpolating models.

| | TIN | IDW | Natural Neighbor | Kriging |
|---|---|---|---|---|
| Field 1 | 0.077 m | 0.083 m | 0.082 m | 0.082 m |
| Field 2 | 0.046 m | 0.053 m | 0.048 m | 0.049 m |

According to the topographic map of the TIN models that were most suitable to depict farmland terrains, cut-fill analysis was conducted for each field. Based on the statistical

results of the ground elevation data from the topographic maps, the average value of each field's ground elevation data was calculated as 63.597 m and 63.564 m. An average value of ground elevation was assigned to new raster models that shared the same geospatial coordinates and resolution with the topographic maps of the TIN model as the desired ground elevation. By comparing the value of ground elevation for each point in the topographic map with the new raster model, operations of cut or fill could be determined, as shown in Figure 13; blue represents the areas where ground elevation is larger than the average value and soils could be cut to form a desired plane, while red represents the areas where ground elevation is smaller than the average value and it needs extra soils to be filled. Furthermore, the results of the cut-fill analysis showed that for Field 1 the volume of cut soil is about 33.2 m$^3$, which accounted for about 616.4 m$^2$ (shown in blue in Figure 13a), while the areas to be filled with extra soils accounted for about 454.8 m$^2$ (shown in red in Figure 13a). On the other hand, for Field 2 the volume of cut soil is about 46.3 m$^3$, which accounted for about 603.8 m$^2$ (showed in blue in Figure 13b), and the areas to be filled with extra soils accounted for about 557.6 m$^2$ (shown in red in Figure 13b).

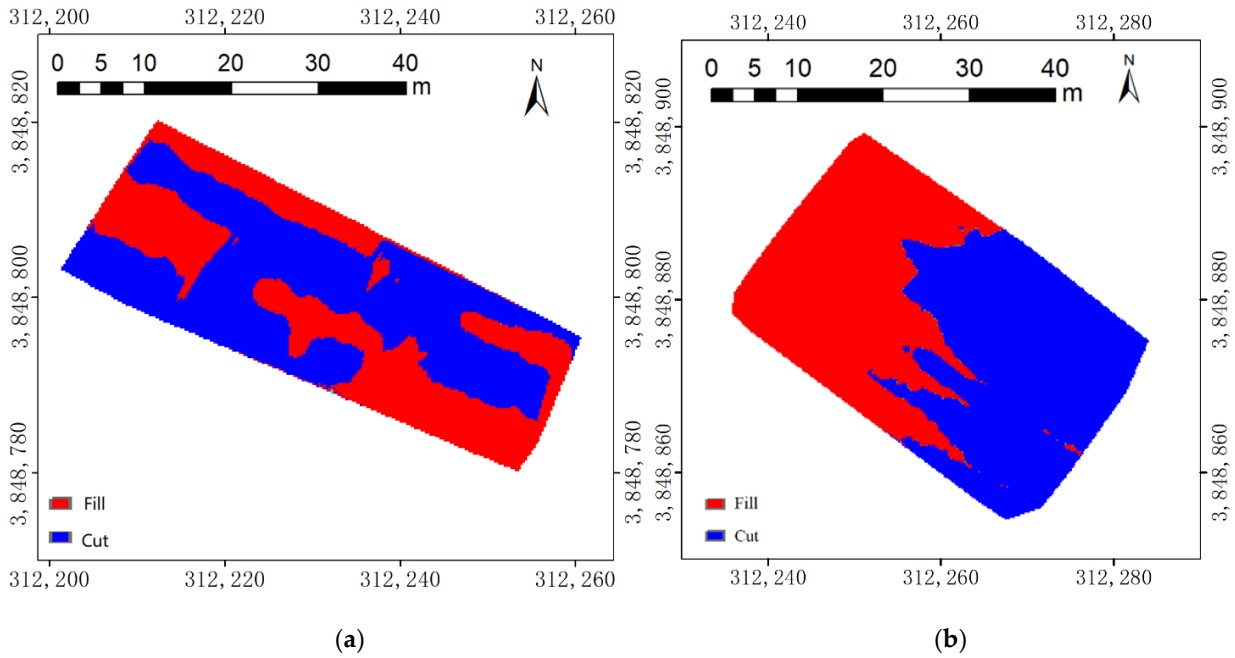

**Figure 13.** Cut-fill analysis based on topographic maps of the TIN model. (**a**) Field 1; (**b**) Field 2.

## 4. Conclusions

In this study, an innovative topographic mapping system integrating LiDAR distance measurements with PPK-GNSS coordinates was developed using a low-altitude UAV as a platform in a simple, efficient, and totally autonomous fashion. The positioning accuracy of the PPK-GNSS module was experimentally validated as 2 cm, which shows a good capability for conducting topographic mapping of farmlands. Topographic mapping experiments in two fields were conducted and topographic maps indicating spatial variations in within-field ground elevation were generated. The main conclusions were summarized as follows.

(1). The RMSE between 12 pairs of ground elevation data of the handheld PPK-GNSS module and UAV-LiDAR topographic mapping system was calculated as 0.041 m and 0.036 m for Field 1 and Field 2, respectively. As the varying range of ground elevation of Field 1 and Field 2 is 0.49 m and 0.363 m, respectively, the RMSE showed that the topographic data of the UAV-LiDAR topographic mapping system are of high accuracy.

(2). TIN proved to be the most suitable interpolation model for applications of farmland topographic mapping. Moreover, as the spatial resolution of topographic mapping points is improved, the interpolating error of topographic maps based on the TIN model

drops drastically from 0.077 m to 0.046 m. Cut-fill analysis based on the topographic maps of TIN model suggested volumes and areas of soils to be cut or filled, which also provides detailed information on setting the height of the desired ground plane and path planning of farmland levels for future study.

Therefore, we can conclude that based on the TIN interpolation model, the UAV-LiDAR topographic mapping system could be successfully used to collect topographic data with high accuracy, which is instructive for precision farmland levelling.

**Author Contributions:** Conceptualization, M.D. and A.R.; methodology, M.D.; investigation, M.D. and H.L.; resources, M.D.; data curation, M.D.; writing—original draft preparation, M.D.; writing— review and editing, M.D., H.L. and A.R.; visualization, M.D.; supervision, M.D.; project administration, M.D.; funding acquisition, M.D. All authors have read and agreed to the published version of the manuscript.

**Funding:** This research was jointly funded by the National Key Research and Development Program of China, grant number 2019YFE0125500 and the Key Scientific Research Projects of Colleges and Universities in Henan Province, grant number 20A416001.

**Institutional Review Board Statement:** Not applicable.

**Informed Consent Statement:** Not applicable.

**Conflicts of Interest:** The authors declare no conflict of interest.

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
