# Peer review of "Design and Experimental Study on an Innovative UAV-LiDAR Topographic Mapping System for Precision Land Levelling"

_drones, doi:10.3390/drones6120403_

Round 1

Reviewer 1 Report

This paper reports the development and experimental study on an innovative approach of generating topographic maps at farmland-level with advantages of high efficiency and simplicity of implementation. In this paper, low-altitude unmanned aerial vehicle (UAV) is used as a platform to integrate laser ranging and PPK-GPS data to obtain cm-level farmland topographic map. The proposed topographic mapping system has good innovation and practicability, and provides guidance information for accurate leveling of farmland. However, the following problems still exist in the experimental process:   

1. In chapter 2.1 Experiment equipment, the specific parameters of the six-rotor UAV and laser ranging module used in the experiment are not given, especially the parameters of flight control module, which will lead to the unknown basis of the designed experimental parameters.

2.  In chapter 2.1 Experiment equipment, autonomous UAV use flight paths designed beforehand at the speed of about 5 m/s and altitude of about 30 m above ground level.

Are there any pre-experimental comparisons or references available to select this speed and height? Are this speed and height related to the performance of the laser ranging module? 

3. (line 121122123) The spatial resolution of the topographic surveying data is determined by the flight speed, positioning frequency of GPS module, and cross-track interval. But what is the positioning frequency of the GPS module? How to calculate the spatial resolution of the topographic surveying data? Is there a requirement for the size of the spatial resolution?

4. In 2.2 Acquiring PPK-GPS coordinates, the figure and data can fully prove the horizontal position accuracy of PPK-GPS module, but it does not describe the accuracy of its vertical position.

5. From Figure 4 we may observe the variations of UAVs flight altitude from 90 m to 92.5 m, which is caused by its intrinsic aerodynamic factors and influence of turbulence. Does this UAVs flight altitude range include PPK-GPS altitude positioning errors?

6. In Chapter 3, is it considered to prove the accuracy of the effective topographic mapping data first (Chapter 3.3) and then to use the effective topographic mapping data to generate the topographic map of the farmland. (Chapter 3.2)?

7. In Conclusion, the first conclusion is not consistent with the topic of the paper, and there is no data in this conclusion to show the topographic map accuracy, so it is considered to omit this conclusion. The third conclusion is similarly lacking in data.

Author Response

Thank you for your valuable comments. The paper has been revised according to your suggestion.

1. Parameters of equipment of UAV-LiDAR topographic mapping system including the UAV platform, GNSS module, and LiDAR were added and listed in Table 1.

2. We determined the flight height as 30 meters mainly for 2 reasons: First, the selection of flight height does have relations with the laser ranging module. Although  the measuring range of the laser ranger could be up to 300 meters, the measuring accuracy deteriorates with the increase of distance. So it is advisable to keep the measuring distance as short as possible. Second,  in consideration for further study on data integration with digtal surface model generated by using aerial images captured by cameras mounted on the drone, the flight height cannot be set too low, orthewise it would be difficult to generate high-quality digital surface model. As for the flight speed of 5 m/s, efficiency and stability were taken into account. When the flight speed exceeds 5 m/s, the drone body tilts too heavily, which affects the stability of the topographic surveying system and accuracy of laser ranging data.

3. The positioning frequency of the GPS module was described in line 111, 112 as 10 Hz, and we listed the parameter in the newly-added Table 1. So the in-track interval of the topographic surveying data can be calculated to be 0.5 m, as the the flight speed (5 m/s) divided by the positioning frequency (10Hz). Since the cross-track interval of field 1 and field 2 was set as about 5 m and 2.5 m, spatial resolution of the topographic surveying data for field 1 and field 2 could be calculated as about 5 m×0.5 m and 2.5 m×0.5 m, described in line 121 to 125. And as for the requirement for the size of the spatial resolution, it is also our reserach interest in the paper, and we concluded that as the spatial resolution of topographic surveying points is intensified from 5 m×0.5 m to 2.5 m×0.5 m, the accuracy of topographic map  improves drastically from 7.7 cm to 4.6 cm. 

4. Technically, the accuracy of vertical position of PPK-GPS coordinates is definately correlated with the horizontal position, the error of which equals about 1 to 2 times of the later. So we examined the accuracy of horizontal position of the PPK-GPS coordinates by convention, and in consideration of the length and conciseness, here we omitted the comparison of vertical position.

5. This UAVs flight altitude range included PPK-GPS altitude positioning errors, which is about 2 to 4 cm.

6. We appreciated your valuable advice, and the accuracy evaluation part of old Chapter 3.3 was rearranged before the old Chapter 3.2 (Generating topographic maps based on interpolation models), which is now renumbered as Chapter 3.3, and we split the other part of old Chapter 3.3 as Chapter 3.4 (Evaluating accuracy of topographic maps).

7. Conclusion (1) and (3) were both deleted.

Thanks again for your value comments.

Reviewer 2 Report

Hi Authors

Please attend to the comments on the attached document. 

Regards

Author Response

Dear reviewer,
Thank you for your valuable comments. The paper has been revised according to your suggestion.

1. The reference was added.
2. The sentences were shortened and referenced, and term of GPS was replaced with GNSS  in this paper.
3. The references were added.
4.  (Digital Surface Model) for the abbreviation of DSM was added.
5. The source was referenced.
6. It was modified as "This study developed...".
7. Modified as "was".
8. The sentence were shortened.
9. Modified as "..., from which the horizontal positioning accuracy of standalone GNSS module can be determined as about 3 m."
10. "respectively" was inserted.
11. This repeated part was deleted.
12. This part was moved to the Results Section as "3.1 Evaluating accuracy of UAV-LiDAR topographic mapping system " 
13. Modified as "was".
14. This sentence was referenced.
15. This part was moved to Material and Method Section as "2.4 Acquiring 3D coordinates of ground surveying points".
16. This part was moved to Material and Method Section as "2.5 Generating topographic maps based on interpolation models".
17. Equations of RMSE was deleted. 
18. "we" was modifed as third person.

Thank you again for your suggestion.

Reviewer 3 Report

Dear authors,
I have reviewed the paper entitled "Design and Experimental Study on an Innovative UAV-LiDAR Topographic Mapping System for Precision Land Levelling". I liked to read your paper. However, there some revisions that should be added to improve the quality of your article:

1- English needs considerable work. There are several grammatical errors, punctuation problems, and many sentences are oddly constructed and confusing. A thorough revision by someone more familiar with the English language is strongly recommended.

2- Did the authors used GPU or CPU to run these algorithms? You need to mention these type of important information.

3- In my point of view, the comparison must be also compared with other authors works (in the same subject). Thus, the authors must create a table regarding to this issue (some frameworks to solve Topographic Mapping System problems using LiDARS: OpenDroneMap, Pix4D, etc... ).

4- The authors should create a Table regarding to the 3D processing time and comparing with other works to validate the proposed algorithm in terms of time.

Author Response

Dear reviewer,

Thank you for your valuable comments. The paper has been revised according to your suggestion.

1. A thorough revision has been made, and English language has been improved.

2. GPU was not used in this paper, as it doesn't incorporate 3D model construction or point cloud processing. In this paper, we acquried PPK-GNSS coordinates, laser ranging data and UAV's attitude data. These data were aligned and processed mostly in EXCEL and MATLAB, which doesn't need high-perfomance computing capability.

3. The related work has been cited as reference 7, 8, 12, 13, 14, 17, and 18. 

4. As explained in the reply of Question 2, this research doesn't include 3D modeling, so the computing time may not not taken into consideration.

Thank you again for your suggestion.

Reviewer 4 Report

Design and experimental study on an innovative UAV-LiDAR topographic mapping system for precision land levelling

Mengmeng Du, Hanyuan Li, and Ali Roshanianfard

The study evaluates the applicability of a novel topographic mapping system for precision land levelling based on UAV-LiDAR. The main goal of the ms is to generate topographic maps at the farmland level with high efficiency and ease of implementation, simplifying a time-consuming operation when carried out at ground level.

The ms deal with an interesting topic providing an innovative approach to address it. The experimental design is well suited with the objectives of the study, and the set of measurements which was taken allows to finely analyze the results obtained in order to evaluate the suitability of the new approach proposed.

I am just skeptical about the applicability of the system in a farmland contest (proposed throughout the text e.g., line 88, 437, 461…) where usually it is needed to face with wide surfaces (we speak about hectares). In this experiment the authors tested the system in small fields (<1500 m2).

The ms is generally well organized and carefully written and the discussion of the data is based on recent literature.

I have a couple of indications which should be implemented to improve the comprehension of the ms:

1.     Figure 1. Please, replace the pictures of the fields with orthophotos in order to make the site's characteristics visible.

2.     Figures 9, 10, 11, 12. Since the elevation differences are so subtle, use the same discrete classification values in all the maps in order to make them comparable between each other.

Therefore, I recommend the article to be accepted for publication, after minor revision.

Author Response

Dear reviewer,

Thank you for your valuable comments. The paper has been revised according to your suggestion.

1. We are sorry but this paper utilizes PPK-GNSS coordinates, laser ranging data and UAV's attitude data to generate topographic maps for farmland, so we don't have orthophotos of the aerial images. As for the field site's characteristics, it was described in text as "Previous crop was maize, and weeds inside the fallow fields were manually removed prior to the experiment, so that there were no significant foreign attachments on ground surface."

2. As you may have suggested, intervals of discrete classification have been set to the same values.

Thank you again for your suggestion.

Round 2

Reviewer 1 Report

1. In chapter 2.1 Experiment equipment, what are the red and green dots in Figure 1? Does the black line indicate the actual flight path of the UAV? It is recommended to add a note in Figure 1.

2.  In chapter 2.1 Experiment equipment, it is recommended to supplement the basis of flight speed (5m/s) and flight height (30m). 

3. (line 119120121) The spatial resolution of the topographic surveying data is determined by the flight speed, positioning frequency of GPS module, and cross-track interval. At the same time, the spatial resolution of the topographic surveying data will affect the accuracy of the system, so it is recommended to add the reasons why the flight speed cannot be reduced.

4. In 2.2 Acquiring PPK-GPS coordinates, considering the effect of PPK-GPS vertical position accuracy on system accuracy, it is suggested to add proof of vertical position accuracy.

Author Response

Dear reviewer,

Thank you for your valuable comments. The paper has been revised according to your suggestion.

  1. Notes  indicating the lines and dots were added in Figure 1.
  2. Descriptions for the flight speed and height were added as "The flight speed was determined out of overall consideration of efficiency and attitude stability of the UAV platform. Low flight speed results in low efficiency, and when the flight speed increases, the pitch angle of UAV body correspondingly changes violently, which affects the stability of the topographic surveying system and accuracy of laser ranging data. From the flight speed and positioning frequency of GNSS module, in-track intervals of the UAV-LiDAR topographic surveying points could be determined as 0.5 m. Cross-track interval of field 1 and field 2 was set as about 5 m and 2.5 m, respectively, to cover the whole field in each flight, according to the endurance limitation of common civilian UAVs. And the influence of spatial resolution of the topographic surveying points on the accuracy of topographic maps by using interpolation models was studied, which is jointly determined by the cross-track interval and the in-track interval. As the in-track interval was already far less than the cross-track interval, the merits of further reducing the flight speed for acquiring denser topographic surveying points was not discussed. Therefore, spatial resolution of the topographic surveying data for field 1 and field 2 was calculated as about 5 m×0.5 m and 2.5 m×0.5 m, respectively, and trajectories of 2 autonomous flights over the experimental fields were shown in Figure 1. Because the altitude of the UAV-LiDAR topographic surveying system affects the accuracy of LiDAR’s distance measurement, it should be set as low as possible. In this study, the altitude was set to 30 m above ground level to conduct accurate topographic surveying and capture aerial images at the same time for further study on data integration."
  3. This issue was also addressed in the answer of the 2nd question.
  4. Figure 7(c) describing the vertical positioning accuracy of PPK-GNSS was added.

Reviewer 3 Report

The authros have revised the manuscript accordingly. I suggest this paper can be accepted now.

Author Response

Dear reviewer,

Thank you again. We do appreciate your valuable comments.